# Using Embeddings to Correct for Unobserved Confounding in Networks

Victor Veitch[1], Yixin Wang[1], and David M. Blei[1,2]

[1]*Department of Statistics, Columbia University*
[2]*Department of Computer Science, Columbia University*

## Abstract

We consider causal inference in the presence of unobserved confounding. We study the case where a proxy is available for the unobserved confounding in the form of a network connecting the units. For example, the link structure of a social network carries information about its members. We show how to effectively use the proxy to do causal inference. The main idea is to reduce the causal estimation problem to a semi-supervised prediction of both the treatments and outcomes. Networks admit high-quality embedding models that can be used for this semi-supervised prediction. We show that the method yields valid inferences under suitable (weak) conditions on the quality of the predictive model. We validate the method with experiments on a semi-synthetic social network dataset. Code at github.com/vveitch/causal-network-embeddings.

## 1 Introduction

We consider causal inference in the presence of unobserved confounding, i.e., where unobserved variables may affect both the treatment and the outcome. We study the case where there is an observed proxy for the unobserved confounders, but (i) the proxy has non-iid structure, and (ii) a well-specified generative model for the data is not available.

**Example 1.1.** We want to infer the efficacy of a drug based on observed outcomes of people who are connected in a social network. Each unit $i$ is a person. The treatment variable $t_i$ indicates whether they took the drug, a response variable $y_i$ indicates their health outcome, and latent confounders $z_i$ might affect the treatment or response. For example, $z_i$ might be unobserved age or sex. We would like to compute the average treatment effect, controlling for these confounds. We assume the social network itself is associated with $z$, e.g., similar people are more likely to be friends. This means that the network itself may implicitly contain confounding information that is not explicitly collected. □

In this example, inference of the causal effect would be straightforward if the confounder $z$ were available. So, intuitively, we would like to infer substitutes for the latent $z_i$ from the underlying social network structure. Once inferred, these estimates $\hat{z}_i$ could be used as a substitute for $z_i$ and we could estimate the causal effect [SM16].

For this strategy to work, however, we need a well-specified generative model (i.e., joint probability distribution) for $z$ and the full network structure. But typically no such model is available. For example, generative models of networks with latent unit structure—such as stochastic block models [WW87; Air+08] or latent space models [Hof+02]—miss properties of real-world networks [Dur06; New09; OR15]. Causal estimates based on substitutes inferred from misspecified models are inherently suspect.

Embedding methods offer an alternative to fully specified generative models. Informally, an embedding method assigns a real-valued embedding vector $\hat{\lambda}_i$ to each unit, with the aim that conditioning on the embedding should decouple the properties of the unit and the network structure. For example, $\hat{\lambda}_i$ might be chosen to explain the local network structure of user $i$.

The embeddings are learned by minimizing an objective function over the network, with no requirement that this objective correspond to any generative model. For pure predictive tasks, e.g., classification of vertices in a graph, embedding-based approaches are state of the art for many real-world datasets [e.g., Per+14; Cha+17; Ham+17; Ham+17; Vei+19a]. This suggests that network embeddings might be usefully adapted to the inference of causal effects.

The method we develop here stems from the following insight. Even if we knew the confounders $\{z_i\}$ we would not actually use all the information they contain to infer the causal effect. Instead, if we use estimator $\hat{\psi}_n$ to estimate the effect $\psi$, then we only require the part of $z_i$ that is actually used by the estimator $\hat{\psi}_n$. For example, if $\hat{\psi}_n$ is an inverse probability weighted estimator [CH08] then we require only estimates for the propensity scores $P(T_i = 1 \mid z_i)$ for each unit.

What this means is that if we can build a good *predictive* model for the treatment then we can plug the outputs into a causal effect estimate directly, without any need to learn the true $z_i$. The same idea applies generally by using a predictive model for both the treatment and outcome. Reducing the causal inference problem to a predictive problem is the crux of this paper. It allows us to replace the assumption of a well-specified model with the more palatable assumption that the black-box embedding method produces a strong predictor.

The contributions of this paper are:

- a procedure for estimating treatment effects using network embeddings;
- an extension of robust estimation results to (non-iid) network data, showing the method yields valid estimates under weak conditions;
- and, an empirical study of the method on social network data.

## 2   Related Work

Our results connect to a number of different areas.

**Causal Inference in Networks.**  Causal inference in networks has attracted significant attention [e.g., SM16; Tch+17; Ogb+17; OV17; Ogb18]. Much of this work is aimed at inferring the causal effects of treatments applied using the network; e.g., social influence or contagion. A major challenge in this area is that homophily—the tendency of similar people to cluster in a network—is generally confounded with contagion—the influence people have on their neighbors [ST11]. In this paper, we assume that each person's treatment and outcome are independent of the network once we know that person's latent attributes; i.e., we assume pure homophily. This is a reasonable assumption in some situations, but certainly not all. Our major motivation is simply that pure homophily is the simplest case, and is thus the natural proving ground for the use of black-box methods in causal network problems. It is an import future direction to extend the results developed here to the contagion case.

Shalizi and McFowland III [SM16] address the homophily/contagion issue with a two-stage estimation procedure. They first estimate latent confounders (node properties), then use these in a regression based estimator in the second stage. Their main result is a proof that if the network was actually generated by either a stochastic block model or a latent space model then the estimation procedure is valid. Our main motivation here is to avoid such well-specified model assumptions. Their work is complementary to our approach: we impose a weaker assumption, but we only address homophily.

**Causal Inference Using Proxy Confounders.**  Another line of connected research deals with causal inference with hidden confounding when there is an observed proxy for the confounder [KM99; Pea12; KP14; Mia+18; Lou+17]. This work assumes the data is generated independently and identically as $(X_i, Z_i, T_i, Y_i) \overset{\text{iid}}{\sim} P$ for some data generating distribution $P$. The variable $Z_i$ causally affects $T_i, Y_i$, and $X_i$. The variable(s) $X_i$ are interpreted as noisy versions of $Z_i$. The main question here is when the causal effect is (non-parametrically) identifiable. The typical flavor of the results is: if the proxy distribution satisfies certain conditions then the marginal distribution $P(Z_i, T_i, Y_i)$ is identifiable, and thus so too is the causal effect (though weaker identification conditions are possible

[Mia+18]). The main differences with the problem we address here are that the network surrogate has non-iid structure, we expect that the information content of the exact confounder can be recovered in the infinite-sample limit, and we do not demand recovery the true data generating distribution.

**Double machine learning.** Chernozhukov et al. [Che+17a] addresses robust estimation of causal effects in the i.i.d. setting. Mathematically, our main estimation result, theorem 5.1, is a fairly straightforward adaptation of their result. The important distinction is conceptual: we treat a different data generating scenario.

**Embedding methods.** Veitch et al. [Vei+19b] use the strategy of reducing causal estimation to prediction to harness text embedding methods for causal inference with text data. In particular, that paper views the embeddings as a dimension reduction strategy and asks how the dimension reduction can be achieved in a manner that preserves causal identification.

## 3 Setup

We first fix some notation and recall some necessary ideas about the statistical estimation of causal effects. We take each statistical unit to be a tuple $O_i = (Y_i, T_i, Z_i)$, where $Y_i$ is the response, $T_i$ is the treatment, and $Z_i$ are (possibly confounding) unobserved attributes of the units. We assume that the units are drawn independently and identically at random from some distribution $P$, i.e., $O_i \overset{\text{iid}}{\sim} P$. We study the case where there is a network connecting the units. We assume that the treatments and outcomes are independent of the network given the latent attributes $\{Z_i\}$. This condition is implied by the (ubiquitous) exchangeable network assumption [OR15; VR15; CD15], though our requirement is weaker than exchangeability.

The average treatment effect of a binary outcome is defined as
$$\psi = \mathbb{E}[Y \mid \mathrm{do}(T = 1)] - \mathbb{E}[Y \mid \mathrm{do}(T = 0)].$$
The use of Pearl's $\mathrm{do}$ notation indicates that the effect of interest is causal: what is the expected outcome if we intervene by assigning the treatment to a given unit? If $Z_i$ contains all common influencers (a.k.a. confounders) of $Y_i$ and $T_i$ then the causal effect is identfiable as a parameter of the observational distribution:
$$\psi = \mathbb{E}[\mathbb{E}[Y \mid Z, T = 1] - \mathbb{E}[Y \mid Z, T = 0]]. \tag{3.1}$$

Before turning to the unobserved $Z$ case, we recall some ideas from the case where $Z$ is observed. Let $Q(t, z) = \mathbb{E}[Y \mid t, z]$ be the conditional expected outcome, and $\hat{Q}_n$ be an estimator for this function. Following 3.1, a natural choice of estimator $\hat{\psi}_n$ is:
$$\hat{\psi}_n^Q = \frac{1}{n} \sum_i \left[ \hat{Q}_n(1, z_i) - \hat{Q}_n(0, z_i) \right].$$

That is, $\psi$ is estimated by a two-stage procedure: First, produce an estimate for $\hat{Q}_n$. Second, plug $\hat{Q}_n$ into a pre-determined statistic to compute the estimate.

Of course, $\hat{\psi}_n^Q$ is not the only possible choice of estimator. In principle, it is possible to do better by incorporating estimates $\hat{g}_n$ of the propensity scores $g(z) = \mathrm{P}(T = 1 \mid z)$. The augmented inverse probability of treatment weighted (A-IPTW) estimator $\hat{\psi}_n^A$ is an important example [Rob+00; Rob00]:
$$\hat{\psi}_n^A = \frac{1}{n} \sum_i \hat{Q}_n(1, z_i) - \hat{Q}_n(0, z_i) + \frac{1}{n} \sum_i \left( \frac{I[t_i = 1]}{\hat{g}_n(z_i)} - \frac{I[t_i = 0]}{1 - \hat{g}_n(z_i)} \right) (y_i - \hat{Q}_n(t_i, z_i)). \tag{3.2}$$

We call $\eta(z) = (Q(0, z), Q(1, z), g(z))$ the nuisance parameters. The main advantage of $\hat{\psi}_n^A$ is that it is robust to misestimation of the nuisance parameters [Rob+94; vR11; Che+17a]. For example, it has the double robustness property: $\hat{\psi}_n$ is consistent if either $\hat{g}_n$ or $\hat{Q}_n$ is consistent. If both are consistent, then $\hat{\psi}_n^A$ is the asymptotically most efficient possible estimator [Bic+00]. We will show below that the good theoretical properties of the suitably modified A-IPTW estimator persist for the embedding method even in the non-iid setting of this paper.

There is a remaining complication. In the general case, if the same data $\boldsymbol{O}_n$ is used to estimate the nuisance parameters $\hat{\eta}_n$ and to compute $\hat{\psi}_n(\boldsymbol{O}_n; \hat{\eta}_n)$ then the estimator is not guaranteed to maintain good asymptotic properties. This problem can be solved by splitting the data, using one part to estimate $\hat{\eta}_n$ and the other to compute the estimate [Che+17a]. We rely on this data splitting approach.

# 4    Estimation

We now return to the setting where the $\{z_i\}$ are unobserved, but a network proxy is available.

Following the previous section, we want to hold out a subset of the units $i \in I_0$ and, for each of these units, produce estimates of the propensity score $g(z_i)$ and the conditional expected outcome $Q(t_i, z_i)$. Our starting point is (an immediate corollary of) [RR83, Thm. 3]:

**Theorem 4.1.** *Suppose $\lambda(z)$ is some function of the latent attributes such that at least one of the following is $\lambda(Z)$-measurable: (i) $(Q(0, Z), Q(1, Z))$, or (ii) $g(Z)$. If adjusting for $Z$ suffices to render the average treatment effect identifiable then adjusting for only $\lambda(Z)$ also suffices. That is,*
$\psi = \mathbb{E}[\mathbb{E}[Y \mid \lambda(Z), T = 1] - \mathbb{E}[Y \mid \lambda(Z), T = 0]]$

The significance of this result is that adjusting for the confounding effect of the latent attributes does not actually require us to recover the latent attributes. Instead, it suffices to recover only the aspects $\lambda(z_i)$ that are relevant for the prediction of the propensity score or conditional expected outcome.

The idea is that we may view network embedding methods as black-box tools for extracting information from the network that is relevant to solving prediction problems. We make use of embedding based semi-supervised prediction models. What this means is that we assign an embedding $\lambda_i \in \mathbb{R}^p$ to each unit, and define predictors $\tilde{Q}(t_i, \lambda_i; \gamma^Q)$ mapping the embedding and treatment to a prediction for $y_i$, and predictor $\tilde{g}(\lambda_i; \gamma^g)$ mapping the embeddings to predictions for $t_i$. In this context, 'semi-supervised' means that when training the model we do not use the labels of units in $I_0$, but we do use all other data—including the proxy structure on units in $I_0$.

An example clarifies the general approach.

**Example 4.2.** We denote the network $G_n$. We assume a continuous valued outcome. Consider the case where $\tilde{Q}(0, \cdot; \gamma^Q)$, $\tilde{Q}(1, \cdot; \gamma^Q)$ and $\text{logit } \tilde{g}(\cdot; \gamma^g)$ are all linear predictors. We train a model with a relational empirical risk minimization procedure [Vei+19a]. We set:

$$\hat{\lambda}_n, \hat{\gamma}_n^Q, \hat{\gamma}_n^g = \underset{\lambda, \gamma^Q, \gamma^g}{\operatorname{argmin}} \mathbb{E}_{G_k = \mathsf{Sample}(G_n, k)}[L(G_k; \lambda, \gamma^Q, \gamma^g)]$$

where $\mathsf{Sample}(G_n, k)$ is a randomized sampling algorithm that returns a random subgraph of size $k$ from $G_n$ (e.g., a random walk with $k$ edges), and

$$L(G_k; \lambda, \gamma^Q, \gamma^g) = \sum_{i \in I \setminus I_0} (y_i - \tilde{Q}(t_i, \lambda_i; \gamma^Q))^2 + \sum_{i \in I \setminus I_0} \mathsf{CrossEntropy}(t_i, \tilde{g}(\lambda_i; \gamma^g))$$
$$+ \sum_{i,j \in I \times I} \mathsf{CrossEntropy}(1[(i,j) \in G_k], \mathsf{sigmoid}(\lambda_i^T \lambda_j)).$$

Here, $I$ is the full set of units, and $1[(i, j) \in G_k]$ indicates whether units $i$ and $j$ are linked. Note that the final term of the model is the one that explains the relational structure. Intuitively, it says that the logit probability of an edge is the inner product of the embeddings of the end points of the edge. This loss term makes use of the entire dataset, including links that involve the heldout units. This is important to ensure that the embeddings for the heldout data 'match' the rest of the embeddings. □

**Estimation.**    With a trained model in hand, computing the estimate of the treatment effect is straightforward. Simply plug-in the estimated values of the nuisance parameters to a standard estimator. For example, using the A-IPTW estimator eq. (3.2),

$$\hat{\psi}_n^{\mathrm{A}}(I_0) := \frac{1}{|I_0|} \sum_{i \in I_0} \tilde{Q}(1, \hat{\lambda}_{n,i}; \hat{\gamma}_n^Q) - \tilde{Q}(0, \hat{\lambda}_{n,i}; \hat{\gamma}_n^Q)$$
$$+ \frac{1}{|I_0|} \sum_{i \in I_0} \left( \frac{I[t_i = 1]}{\tilde{g}(\hat{\lambda}_{n,i}; \hat{\gamma}_n^g)} - \frac{I[t_i = 0]}{1 - \tilde{g}(\hat{\lambda}_{n,i}; \hat{\gamma}_n^g)} \right) (y_i - \tilde{Q}(t_i, \hat{\lambda}_{n,i}; \hat{\gamma}_n^Q)). \tag{4.1}$$

We also allow for a more sophisticated variant. We split the data into $K$ folds $I_0, \dots, I_{K-1}$ and define our estimator as:

$$\hat{\psi}_n^{\mathrm{A}} = \frac{1}{K} \sum_j \hat{\psi}_n^{\mathrm{A}}(I_j). \tag{4.2}$$

This variant is more data efficient than just using a single fold. Finally, the same procedure applies to estimators other than the A-IPTW. We consider the effect of the choice of estimator in section 6.

# 5  Validity

When does the procedure outlined in the previous section yield valid inferences? We now present a theorem establishing sufficient conditions. The result is an adaption of the "double machine learning" of Chernozhukov et al. [Che+17a; Che+17b] to the network setting. We first give the technical statement, and then discuss its significance and interpretation.

Fix notation as in the previous section. We also define $\hat{\gamma}_n^{Q,I_k^c}$ and $\hat{\gamma}_n^{g,I_k^c}$ to be the estimates for $\gamma_Q, \gamma_g$ calculated using all but the $k$th data fold.

**Assumption 1.** The probability distributions $P$ satisfies

$$Y = Q(T, Z) + \zeta, \qquad \mathbb{E}[\zeta \mid Z, T] = 0,$$
$$T = g(Z) + \nu, \qquad \mathbb{E}[\nu \mid Z] = 0.$$

Further, we requrie that $T$ does not causally affect either $Z$ or the network.

The second part of the statement is necessary to rule out a linear-gaussian edge case.

**Assumption 2.** There is some function $\lambda$ mapping features $Z$ into $\mathbb{R}^p$ such that $\lambda$ satisfies the condition of theorem 4.1, and each of $||\tilde{Q}_n(0, \hat{\lambda}_{n,i}; \hat{\gamma}_{Q,I_k^c}) - Q(0, \lambda(Z_i))||_{P,2}$, $||\tilde{Q}_n(1, \hat{\lambda}_{n,i}; \hat{\gamma}_{Q,I_k^c}) - Q(1, \lambda(Z_i))||_{P,2}$, and $||\tilde{g}_n(\hat{\lambda}_{n,i}; \hat{\gamma}_{g,I_k^c}) - g(\lambda(Z_i))||_{P,2}$ goes to 0 as $n \to \infty$. Additionally, $\lambda$ must satisfy all of the following assumptions.

**Assumption 3.** The following moment conditions hold for some fixed $\varepsilon, C, c$, some $q > 4$, and all $t \in \{0, 1\}$

$$||Q(t, \lambda(Z))||_{P,q} \leq C,$$
$$||Y||_{P,q} \leq C,$$
$$P(\varepsilon \leq g(\lambda(Z)) \leq 1 - \varepsilon) = 1,$$
$$P(\mathbb{E}_P\left[\zeta^2 \mid \lambda(Z)\right] \leq C) = 1,$$
$$||\zeta||_{P,2} \geq c,$$
$$||\nu||_{P,2} \geq c.$$

**Assumption 4.** The estimators of nuisance parameters satisfy the following accuracy requirements. There is some $\delta_n, \Delta_{n_K} \to 0$ such that for all $n \geq 2K$ and $d \in \{0, 1\}$ it holds with probability no less than $1 - \Delta_{n_K}$:

$$||\tilde{Q}_n(d, \hat{\lambda}_{n,i}; \hat{\gamma}_{Q,I_k^c}) - Q(d, \lambda(Z_i))||_{P,2} \cdot ||\tilde{g}_n(\hat{\lambda}_{n,i}; \hat{\gamma}_{g,I_k^c}) - g(\lambda(Z_i))||_{P,2} \leq \delta_{n_K} \cdot n_K^{-1/2} \quad (5.1)$$

And,

$$P(\varepsilon \leq \tilde{g}_n(\hat{\lambda}_{n,i}; \hat{\gamma}_{g,I_k^c}) \leq 1 - \varepsilon) = 1, \tag{5.2}$$

**Assumption 5.** We assume the dependence between the trained embeddings is not too strong: For any $i, j$ and all bounded continuous functions $f$ with mean 0,

$$\mathbb{E}\left[f(\hat{\lambda}_{n,i}) \cdot f(\hat{\lambda}_{n,j})\right] = o(\frac{1}{n}). \tag{5.3}$$

**Theorem 5.1.** *Denote the true ATE as $\psi$. Let $\hat{\psi}_n$ be the $K$-fold A-IPTW variant defined in eq. (4.2). Under Assumptions 1 to 5, $\hat{\psi}_n$ concentrates around $\psi$ with the rate $1/\sqrt{n}$ and is approximately unbiased and normally distributed:*

$$\sigma^{-1}\sqrt{n}(\hat{\psi}_n - \psi) \xrightarrow{d} \mathcal{N}(0, 1)$$
$$\sigma^2 = \mathbb{E}_P\left[\varphi_0^2(Y, T, \lambda(Z); \theta_0, \eta(\lambda(Z)))\right],$$

*where*

$$\varphi_0(Y, T, \lambda(Z); \theta_0, \eta(\lambda(Z))) = \frac{T}{g(\lambda(Z))}\{Y - Q(1, \lambda(Z))\} - \frac{1 - T}{1 - g(\lambda(Z))}\{Y - Q(0, \lambda(Z))\}$$
$$+ \{Q(1, \lambda(Z)) - Q(0, \lambda(Z))\} - \psi.$$

*Proof.* The proof follows Chernozhukov et al. [Che+17b]. The main changes are technical modifications exploiting Assumption 5 to allow for the use of the full data in the embedding training. We defer the proof to the appendix. □

**Interpretation and Significance.** Under suitable conditions, theorem 5.1 promises us that the treatment effect is identifiable and can be estimated at a fast rate. It is not surprising that there are some conditions under which this holds. The insight from theorem 5.1 lies with the particular assumptions that are required.

Assumptions 1 and 3 are standard conditions. Assumption 1 posits a causal model that (i) restricts the treatments and outcomes to a pure unit effect (i.e., it forbids contagion effects), and that (ii) renders the causal effects identifiable when $Z$ observed. Assumption 3 is technical conditions on the data generating distribution. This assumption includes the standard positivity condition. Possible violations of these conditions are important and must be considered carefully in practice. However, such considerations are standard, independent of the non-iid, no-generative-model setting that is our focus, so we do not comment further.

Our first deviation from the standard causal inference setup is Assumption 2. This is the identification condition when $Z$ is not observed. It requires that the learned embeddings are able to extract whatever information is relevant to the prediction of the treatment and outcome. This assumption is the crux of the method.

A more standard assumption would directly posit the relationship between $Z$ and the proxy network; e.g., by assuming a stochastic block model or latent space model. The practitioner is then required to assess whether the posited model is realistic. In practice, all generative models of networks fail to capture the structure of real-world networks. Instead, we ask the practitioner to judge the plausibility of the *predictive* embedding model. Such judgments are non-falsifiable, and must be based on experience with the methods and trials on semi-synthetic data. This is a difficult task, but the assumption is at least not violated a priori.

In practice, we do not expect the identification assumption to hold exactly. Instead, the hope is that applying the method will adjust for whatever confounding information is present in the network. This is useful even if there is confounding exogenous to the network. We study the behavior of the method in the presence of exogenous confounding in section 6.

The condition in Assumption 4 addresses the *statistical* quality of the nuisance parameter estimation procedure. For an estimator to be useful, it must produce accurate estimates with a reasonable amount of data. It is intuitive that if accurately estimating the nuisance parameters requires an enormous amount of data, then so too will estimation of $\psi$. eq. (5.1) shows that this is not so. It suffices, in principle, to estimate the nuisance parameters crudely, e.g., a rate of $o(n^{1/4})$ each. This is important because the need to estimate the embeddings may rule out parametric-rate convergence of the nuisance parameters; theorem 5.1 shows this is not damning.

Assumption 5 is the price we pay for training the embeddings with the full data. If the pairwise dependence between the learned embeddings is very strong then the data splitting procedure does not guarantee that the estimate is valid. However, the condition is weak and holds empirically. The condition can also be removed by a two-stage procedure where the embeddings are trained in an unsupervised manner and then used as a direct surrogate for the confounders. However, such approaches have relatively poor predictive performance [Yan+16; Vei+19a]. We compare to the two-stage approach in section 6.

# 6 Experiments

The main remaining questions are: Is the method able to adjust for confounding in practice? If so, is the joint training of embeddings and classifier important? And, what is the best choice of plug-in estimator for the second stage of the procedure? Additionally, what happens in the (realistic) case that the network does not carry all confounding information?

We investigate these questions with experiments on a semi-synthetic network dataset.[1] We find that in realistic situations, the network adjustment improves the estimation of the average treatment

effect. The estimate is closer to the truth than estimates from either a parametric baseline, or a two-stage embedding procedure. Further, we find that network adjustment improves estimation quality even in the presence of confounding that is exogenous to the network. That is, the method still helps even when full identification is not possible. Finally, as predicted by theory, we find that the robust estimators are best when the theoretical assumptions hold. However, the simple conditional-outcome-only estimator has better performance in the presence of significant exogenous confounding.

**Choice of estimator.** We consider 4 options for the plug-in treatment effect estimator.

1. The conditional expected outcome based estimator,

$$\hat{\psi}_n^Q = \frac{1}{n} \sum_i \left[ \tilde{Q}_n(1, \hat{\lambda}_{n,i}; \hat{\gamma}_n) - \tilde{Q}_n(0, \hat{\lambda}_{n,i}; \hat{\gamma}_n) \right],$$

   which only makes use of the outcome model.
2. The inverse probability of treatment weighted estimator,

$$\hat{\psi}_n^g = \frac{1}{n} \sum_i \left[ \frac{1[t_i = 1]}{\tilde{g}(\hat{\lambda}_{n,i}; \hat{\gamma}_n)} - \frac{1[t_i = 0]}{1 - \tilde{g}(\hat{\lambda}_{n,i}; \hat{\gamma}_n)} \right] Y_i,$$

   which only makes use of the treatment model.
3. The augmented inverse probability treatment estimator $\hat{\psi}_n^A$, defined in eq. (4.1).
4. A targeted minimum loss based estimator (TMLE) [vR11].

The later two estimators both make full use of the nuisance parameter estimates. The TMLE also admits the asymptotic guarantees of theorem 5.1 (though we only state the theorem for the simpler A-IPTW estimator). The TMLE is a variant designed for better finite sample performance.

**Pokec.** To study the properties of the procedure, we generate semi-synthetic data using a real-world social network. We use a subset of the Pokec social network. Pokec is the most popular online social network in Slovakia. For our purposes, the main advantages of Pokec are: the anonymized data are freely and openly available [TZ12; LK14] [2], and the data includes significant attribute information for the users, which is necessary for our simulations. We pre-process the data to restrict to three districts (Žilina, Cadca, Namestovo), all within the same region (Žilinský). The pre-processed network has 79 thousand users connected by 1.3 million links.

**Simulation.** We make use of three user level attributes in our simulations: the `district` they live in, the user's `age`, and their Pokec `join date`. These attributes were selected because they have low missingness and have some dependency with the the network structure. We discretize age and join date to a 3-level categorical variable (to match district).

For the simulation, we take each of these attributes to be the hidden confounder. We will attempt to adjust for the confounding using the Pokec network. We take the probability of treatment to be wholly determined by the confounder $z$, with the three levels corresponding to $g(z) \in \{0.15, 0.5, 0.85\}$. The treatment and outcome for user $i$ is simulated from their confounding attribute $z_i$ as:

$$t_i = \text{Bern}(g(z_i)), \tag{6.1}$$
$$y_i = t_i + \beta(g(z_i) - 0.5) + \varepsilon_i \qquad \varepsilon_i \sim N(0, 1). \tag{6.2}$$

In each case, the true treatment effect is 1.0. The parameter $\beta$ controls the amount of confounding.

**Estimation.** For each simulated dataset, we estimate the nuisance parameters using the procedure described in section 4 with $K = 10$ folds. We use a random-walk sampler with negative sampling with the default relational ERM settings [Vei+19a]. We pre-train the embeddings using the unsupervised objective only, run until convergence.

**Baselines.** We consider three baselines. The first is the naive estimate that does not attempt to control for confounding; i.e., $\frac{1}{m} \sum_{i:t_i=1} y_i - \frac{1}{n-m} \sum_{i:t_i=0} y_i$, where $m$ is the number of treated individuals. The second baseline is the two-stage procedure, where we first train the embeddings on the unsupervised objective, freeze them, and then use them as features for the same predictor maps. The final baseline is a parametric approach to controlling for the confounding. We fit a

**Table 1:** Adjusting using the network improves ATE estimate in all cases. Further, the single-stage method is more accurate than baselines. Table entries are estimated ATE with 10-fold std. Ground truth is 1.0. Low and high confounding correspond to $\beta = 1.0$ and 10.0.

| Conf. | age Low | age High | district Low | district High | join date Low | join date High |
|---|---|---|---|---|---|---|
| Unadjusted | $1.32 \pm 0.02$ | $4.34 \pm 0.05$ | $1.34 \pm 0.03$ | $4.51 \pm 0.05$ | $1.29 \pm 0.03$ | $4.03 \pm 0.06$ |
| Parametric | $1.30 \pm 0.00$ | $4.06 \pm 0.01$ | $1.21 \pm 0.00$ | $3.22 \pm 0.01$ | $1.26 \pm 0.00$ | $3.73 \pm 0.01$ |
| Two-stage | $1.33 \pm 0.02$ | $4.55 \pm 0.05$ | $1.34 \pm 0.02$ | $4.55 \pm 0.05$ | $1.30 \pm 0.03$ | $4.16 \pm 0.06$ |
| $\hat{\psi}_n^{\mathrm{A}}$ | $1.24 \pm 0.04$ | $3.40 \pm 0.04$ | $1.09 \pm 0.02$ | $2.03 \pm 0.07$ | $1.21 \pm 0.05$ | $3.26 \pm 0.09$ |

**Table 2:** The conditional-outcome-only estimator is usually most accurate. Table entries are estimated ATE with 10-fold std. Ground truth is 1.0. Low and high confounding correspond to $\beta = 1.0$ and 10.0.

| Conf. | age Low | age High | district Low | district High | join date Low | join date High |
|---|---|---|---|---|---|---|
| $\hat{\psi}_n^{Q}$ | $1.05 \pm 0.24$ | $2.77 \pm 0.35$ | $1.03 \pm 0.25$ | $1.75 \pm 0.20$ | $1.17 \pm 0.35$ | $2.41 \pm 0.45$ |
| $\hat{\psi}_n^{g}$ | $1.27 \pm 0.03$ | $3.12 \pm 0.06$ | $1.10 \pm 0.03$ | $1.66 \pm 0.07$ | $1.29 \pm 0.05$ | $3.10 \pm 0.07$ |
| $\hat{\psi}_n^{\mathrm{A}}$ | $1.24 \pm 0.04$ | $3.40 \pm 0.04$ | $1.09 \pm 0.02$ | $2.03 \pm 0.07$ | $1.21 \pm 0.05$ | $3.26 \pm 0.09$ |
| $\hat{\psi}_n^{\mathrm{TMLE}}$ | $1.21 \pm 0.03$ | $3.26 \pm 0.07$ | $1.09 \pm 0.04$ | $2.02 \pm 0.05$ | $1.20 \pm 0.05$ | $3.13 \pm 0.09$ |

mixed-membership stochastic block model [GB13] to the data, with 128 communities (chosen to match the embedding dimension). We predict the outcome using a linear regression of the outcome on the community identities and the treatment. The estimated treatment effect is the coefficient of the treatment.

## 6.1 Results

**Comparison to baselines.** We report comparisons to the baselines in table 1. As expected, adjusting for the network improves estimation in every case. Further, the one-stage embedding procedure is more accurate than baselines.

**Choice of estimator.** We report comparisons of downstream estimators in table 2. The conditional-outcome-only estimator usually yields the best estimates, substantially improving on either robust method. This is likely because the network does not carry all information about the confounding factors, violating one of our assumptions. We expect that `district` has the strongest dependence with the network, and we see best performance for this attribute. Poor performance of robust estimators when assumptions are violated has been observed in other contexts [KS07].

**Confounding exogenous to the network.** In practice, the network may not carry information about all sources of confounding. For instance, in our simulation, the confounders may not be wholly predictable from the network structure. We study the effect of exogenous confounding by a second simulation where the confounder consists of a part that can be fully inferred from the network and part that is wholly exogenous.

For the inferrable part, we use the estimated propensity scores $\{\hat{g}_i\}$ from the `district` experiment above. By construction, the network carries all information about each $\hat{g}_i$. We define the (ground truth) propensity score for our new simulation as $\mathrm{logit}\, g_{\mathrm{sim}} = (1-p)\,\mathrm{logit}\,\hat{g}_i + p\xi_i,$

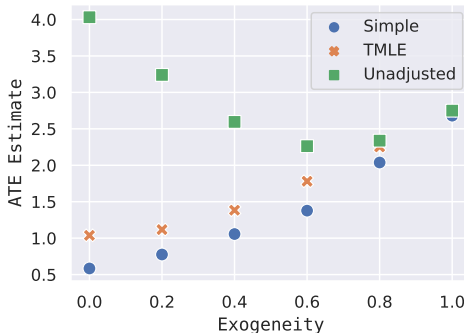

**Figure 1:** Adjusting for the network helps even when the no exogenous confounding assumption is violated. The robust TMLE estimator is the best estimator when no assumptions are violated. The simple conditional-outcome-only estimator ("Simple") is better in the presence of moderate exogeneity. Plot shows estimates of ATE from district simulation. Ground truth is 1.

with $\xi_i \overset{\text{iid}}{\sim} N(0, 1)$. The second term, $\xi_i$, is the exogenous part of the confounding. The parameter $p$ controls the level of exogeneity. We simulate treatments and outcomes as in eq. (6.1).

In fig. 1 we plot the estimates at various levels of exogeneity. We observe that network adjustment helps even when the no exogenous confounding assumption is violated. Further, we see that the robust estimator has better performance when $p = 0$, i.e., when the assumptions of theorem 5.1 are satisfied. However, the conditional-outcome-only estimator is better with substantial exogenous confounding.

## 7 Discussion

We have seen how black-box embedding methods can be harnessed for causal inference in the context of networks. The important conceptual points of the development are: First, the method eliminates the need to precisely specify the properties that influence both network formation and that are confounding. In particular, we need not specify a parametric model for how the network is formed. And, second, identification and estimation can be achieved even if the embedding method extracts the necessary information at only a slow rate. That is, absence of a parametric model is not a grevious problem from a sample-complexity perspective. These are substantial strengths. However, there also significant limitations and opportunity for future work.

Assumption 2 may be difficult to reason about in practice. It requires the practitioner to assess both whether (1) the network carries sufficient information for identification, and (2) the embedding method is able to effectively extract this information. The first part is an assessment based on application-specific domain knowledge. The second part is based on past performance of embedding methodse.g., a method that reliably predicts political affiliation in datasets where affiliation is labeled can be expected to effectively extract information relevant to political identity. This is an improvement on the (impossible) requirement of finding a well-specified model describing how the network was generated. While we do not expect that either condition is exactly satisfied, the exogeneous-confounding experiments in section 6 suggest that applying the adjustment can still improve estimation. An important direction for future work is to develop new methods for sensitivity analysis—applicable in this black-box setting—and formal results about when Assumption 2 can be expected to hold. As an example of the need for such results, partial adjustment for confounding is known to hurt estimation in certain cases [e.g., bias amplification Mid+16; Din+17].

The pure-homophily assumption is restrictive. Many of the most interesting causal questions on networks are explicitly about influence and contagion. We restricted to the homophily case here for simplicity, but it does not appear to be fundamentally required. Extending the results to handle contagion and influence is an important direction for future work.

## Footnotes

[1]Code and pre-processed data at github.com/vveitch/causal-network-embeddings

[2] snap.stanford.edu/data/soc-Pokec.html

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
