[Supplementary Material · proof_of_main_result.pdf]

## A  Proof of Main Result

We now give the proof of Theorem 5.1, which establishes identifiability, consistency, and asymptotic normality.

Recall our setup:

- $Y$: outcome; $T$: treatment; $Z$: confounder.
- $Z$ is unobserved. We use some non-iid additional structure as a proxy.
- $(Y_i, T_i, Z_i) \overset{\text{iid}}{\sim} P$.
- $Q(t, z) = \mathbb{E}\left[Y \mid t, z\right]$; $g(Z) = P(T = 1 \mid Z)$
- The target parameter is the ATE,

$$\psi_0 = \mathbb{E}\left[Q(1, Z) - Q(0, Z)\right].$$

**The estimator and the algorithm.**  Recall that we learn the nuisance parameters $Q$, $g$, and the embeddings $\lambda$ using a semi-supervised embedding-based predictor. We allow a slightly more general construction of the estimator than in the body of the paper. In the body, we state the result only for the A-IPTW. Here, we allow any estimator that solves the efficient estimating equations. This allows, for example, for targeted minimum loss based estimation.

Step 1. Form a $K$-fold partition; the splits are $I_k, k = 1, \ldots, K$. For each set $I_k$, let $I_k^c$ denote the units not in $I_k$.

Construct $K$ estimators $\check{\psi}(I_k^c), k = 1, \ldots, K$:

1. Estimate the nuisance parameters $Q$, $g$, and the embedding $\lambda$:

$$\hat{\eta}(I_k^c) := \left(\hat{\lambda}_i, \tilde{g}_n(\cdot; \hat{\gamma}_n^{g, I_k^c}), \tilde{Q}_n(\cdot, \cdot; \hat{\gamma}_n^{Q, I_k^c})\right)$$

2. $\check{\psi}(I_k^c)$ is a solution to the following equation:

$$\frac{1}{n_K} \sum_{i \in I_k} \varphi\left(Y_i, T_i, Z_i; \psi_0, \hat{\lambda}_i, \tilde{g}_n(\cdot; \hat{\lambda}_i, \hat{\gamma}_n^{g, I_k^c}), \tilde{Q}_n(\cdot, \cdot; \hat{\gamma}_n^{Q, I_k^c})\right) = 0,$$

where the $\varphi(\cdot)$ function is the efficient score:

$$\begin{aligned}
&\varphi(Y, T, Z; \psi_0, \lambda, \tilde{g}_n, \tilde{Q}_n) \\
&= \frac{T}{\tilde{g}_n(\lambda)}\{Y - \tilde{Q}_n(1, \lambda)\} - \frac{1 - T}{1 - \tilde{g}_n(\lambda)}\{Y - \tilde{Q}_n(0, \lambda)\} + \{\tilde{Q}_n(1, \lambda) - \tilde{Q}_n(0, \lambda)\} - \psi_0.
\end{aligned}$$

We note that $\varphi$ does not depend on the unobserved $Z$.

Step 2. The final estimator for the ATE $\psi_0$ is

$$\tilde{\psi} = \frac{1}{K} \sum_{k=1}^{K} \check{\psi}(I_k^c).$$

**The theorem and the proof.**

**Assumption 1.**  The probability distributions $P$ satisfies

$$\begin{aligned}
Y &= Q(T, Z) + \zeta, & \mathbb{E}[\zeta \mid Z, T] &= 0, \\
T &= g(Z) + \nu, & \mathbb{E}[\nu \mid Z] &= 0.
\end{aligned}$$

**Assumption 2.**  There is some function $\lambda$ mapping features $Z$ into $\mathbb{R}^p$ such that $\lambda$ satisfies the condition of Theorem 4.1, and

$$||\tilde{Q}_n(d, \hat{\lambda}_{n,i}; \hat{\gamma}_{Q, I_k^c}) - Q(d, \lambda(Z_i))||_{P,2} + ||\tilde{g}_n(\hat{\lambda}_{n,i}; \hat{\gamma}_{g, I_k^c}) - g(\lambda(Z_i))||_{P,2} \leq \delta_{n_K}. \qquad (5.1)$$

Additionally, $\lambda$ must satisfy all of the following assumptions.

**Assumption 3.** The following moment conditions hold for some fixed $\varepsilon, C, c$, some $q > 4$, and all $t \in \{0, 1\}$

$$||Q(t, \lambda(Z))||_{P,q} \leq C,$$
$$||Y||_{P,q} \leq C,$$
$$P(\varepsilon \leq g(\lambda(Z)) \leq 1 - \varepsilon) = 1,$$
$$P(\mathbb{E}_P\left[\zeta^2 \,|\, \lambda(Z)\right] \leq C) = 1,$$
$$||\zeta||_{P,2} \geq c,$$
$$||\nu||_{P,2} \geq c.$$

**Assumption 4.** The estimators of nuisance parameters satisfy the following accuracy requirements. There is some $\delta_n, \Delta_{n_K} \to 0$ such that for all $n \geq 2K$ and $d \in \{0, 1\}$ it holds with probability no less than $1 - \Delta_{n_K}$:

$$||\tilde{Q}_n(d, \hat{\lambda}_{n,i}; \hat{\gamma}_{Q,I_k^c}) - Q(d, \lambda(Z_i))||_{P,2} \cdot ||\tilde{g}_n(\hat{\lambda}_{n,i}; \hat{\gamma}_{g,I_k^c}) - g(\lambda(Z_i))||_{P,2} \leq \delta_{n_K} \cdot n_K^{-1/2} \quad (5.2)$$

And,

$$P(\varepsilon \leq \tilde{g}_n(\hat{\lambda}_{n,i}; \hat{\gamma}_{g,I_k^c}) \leq 1 - \varepsilon) = 1, \qquad (5.3)$$

**Assumption 5.** We assume the dependence between the trained embeddings is not too strong: For any $i, j$ and all bounded continuous functions $f$ with mean 0,

$$\mathbb{E}\left[f(\hat{\lambda}_{n,i}) \cdot f(\hat{\lambda}_{n,j})\right] = o(\frac{1}{n}). \qquad (5.4)$$

**Theorem A.1** (Validity). *Denote the true ATE as*

$$\psi_0 = \mathbb{E}_P\left[Q(1, Z) - Q(0, Z)\right].$$

*Under Assumptions 1 to 5 the estimator $\tilde{\psi}$ concentrates around $\psi_0$ with the rate $1/\sqrt{n}$ and is approximately unbiased and normally distributed:*

$$\sigma^{-1}\sqrt{n}(\tilde{\psi} - \psi_0) \xrightarrow{d} \mathcal{N}(0, 1)$$
$$\sigma^2 = \mathbb{E}_P\left[\varphi_0^2\left(W; \psi_0, \eta(\lambda(Z))\right)\right],$$

*where*

$$W = (Y, T, \lambda(Z)),$$
$$\eta(\lambda(Z)) = (g(\lambda(Z)), Q(T, \lambda(Z))),$$

*and*

$$\varphi_0(Y, T, \lambda(Z); \psi_0, \eta(\lambda(Z)))$$
$$= \frac{T}{g(\lambda(Z))}\{Y - Q(1, \lambda(Z))\} - \frac{1 - T}{1 - g(\lambda(Z))}\{Y - Q(0, \lambda(Z))\} + \{Q(1, \lambda(Z)) - Q(0, \lambda(Z))\} - \psi_0.$$

*Proof.* We prove the result for the special case where $\lambda$ is the identity map. By Assumption 2 this is without loss of generality—it's the case where all of the information in $Z$ is relevant for prediction. This is not an important mathematical point, but substantially simplifies notation.

The proof follow the same idea as in Chernozhukov *et al.* [Che+17b] with a few modifications accounting for the non-iid proxy structure.

We start with some notation.

1. $|| \cdot ||_{P,q}$ denotes the $L_q(P)$ norm. For example, for measurable $f : \mathcal{W} \xrightarrow{d} \mathbb{R}$,

$$||f(W)||_{P,q} := (\int |f(w)^q \, \mathrm{d}P(w)|)^{1/q}.$$

2. The empirical process $\mathbb{G}_{n,I}(f(W))$ for $||f(W_i)||_{P,2} < \infty$ is

$$\mathbb{G}_{n,I}(f(W)) := \frac{1}{\sqrt{n}} \sum_{i \in I}(f(W_i) - \int f(w) \, \mathrm{d}P(w)).$$

444     3. The empirical expectation and probability is

$$\mathbb{E}_{n,I}\left[f(W)\right] := \frac{1}{n}f(W_i); \qquad \mathbb{P}_{n,I}(A) := \frac{1}{n}\sum_{i\in I}1(W_i\in A).$$

445    Let $\mathbb{P}_n$ be the empirical measure.

446    Step 1: (Main Step). Letting $\check{\psi}_k = \check{\psi}(I_k^c)$, we first write

$$\sqrt{n}(\check{\psi}_k - \psi_0) = \mathbb{G}_{n,I_k^c}\varphi(W;\psi_0,\hat{\eta}(I_k^c)) + \sqrt{n}\int \varphi(w;\psi_0,\hat{\eta}(I_k^c))\,\mathrm{d}\mathbb{P}_n(w), \qquad (\text{A.1})$$

447    where

$$\hat{\eta}(I_k^c) := \left(\hat{\lambda}_i, \tilde{g}_n(\cdot;\hat{\gamma}_n^{g,I_k^c}), \tilde{Q}_n(\cdot,\cdot;\hat{\gamma}_n^{Q,I_k^c})\right)$$

448    as is defined earlier.

449    Steps 2 and 3 below demonstrate that for each $k = 1,\ldots,K$,

$$\int (\varphi(w;\psi_0,\hat{\eta}(I_k^c)) - \varphi_0(w;\psi_0,\eta(z)))^2\,\mathrm{d}\mathbb{P}_n(w) = o_{\mathbb{P}_n}(1), \qquad (\text{A.2})$$

450    and that

$$\sqrt{n}\int \varphi(w;\psi_0,\hat{\eta}(I_k^c))\,\mathrm{d}\mathbb{P}_n(w) = o_{\mathbb{P}_n}(1). \qquad (\text{A.3})$$

451    (A.2) implies

$$\mathbb{G}_{n,I_k^c}\left(\varphi(w;\psi_0,\hat{\eta}(I_k^c)) - \varphi_0(w;\psi_0,\eta(z))\right) = o_{\mathbb{P}_n}(1)$$

452    due to Lemma B.1 of Chernozhukov *et al.* [Che+17b] and the Chebychev's inequality.

453    We note that $\hat{\eta}(I_k^c) = \left(\hat{\lambda}_i, \tilde{g}_n(\cdot;\hat{\gamma}_n^{g,I_k^c}), \tilde{Q}_n(\cdot,\cdot;\hat{\gamma}_n^{Q,I_k^c})\right)$, where the embedding $\hat{\lambda}_i$'s are not indepen-
454    dent. By contrast, $\eta(z)$ only depends on $Z_i$ where all $Z_i$'s are independent.

455    We next show $\sigma^{-1}\sqrt{n_K}(\check{\psi}_k - \psi_0)_{k=1}^K = \sigma^{-1}\mathbb{G}_{n,I_k^c}\varphi_0(W;\psi_0,\eta(Z))_{k=1}^K + o_{\mathbb{P}_n}(1)$.

456    First, we notice

$$\mathbb{E}\left[[\sqrt{n_K}(\check{\psi}_k - \psi_0) - \mathbb{G}_{n,I_k^c}\varphi_0(W;\psi_0,\eta(Z))]^2\,|\,I_k^c\right]$$
$$=\mathbb{E}\left[[\mathbb{G}_{n,I_k^c}\varphi(W;\psi_0,\hat{\eta}(I_k^c)) - \mathbb{G}_{n,I_k^c}\varphi_0(W;\psi_0,\eta(Z)) + o_{\mathbb{P}_n}(1)]^2\,|\,I_k^c\right]$$
$$=\mathbb{E}\left[(\mathbb{G}_{n,I_k^c}\varphi(W;\psi_0,\hat{\eta}(I_k^c)))^2\,|\,I_k^c\right] + \mathbb{E}\left[(\mathbb{G}_{n,I_k^c}\varphi_0(W;\psi_0,\eta(Z)))^2\,|\,I_k^c\right]$$
$$\quad - 2\mathbb{E}\left[(\mathbb{G}_{n,I_k^c}\varphi(W;\psi_0,\hat{\eta}(I_k^c)))\cdot(\mathbb{G}_{n,I_k^c}\varphi_0(W;\psi_0,\eta(Z)))\,|\,I_k^c\right] + o_{\mathbb{P}_n}(1)$$

457    The first equality is due to (A.1) and (A.2). The second equality is due to

$$\mathbb{E}\left[\mathbb{G}_{n,I_k^c}\varphi(W;\psi_0,\hat{\eta}(I_k^c))\right] = \mathbb{E}\left[\mathbb{G}_{n,I_k^c}\varphi_0(W;\psi_0,\eta(Z))\right] = 0. \qquad (\text{A.4})$$

458    If we write $\bar{\varphi}(W_i) := \varphi(W_i) - \int \varphi(w)\,\mathrm{d}\mathbb{P}_n(w)$, we have

$$\mathbb{E}\left[[\sqrt{n_K}(\check{\psi}_k - \psi_0) - \mathbb{G}_{n,I_k^c}\varphi_0(W;\psi_0,\eta(Z))]^2\,|\,I_k^c\right]$$
$$=\frac{1}{n}\mathbb{E}\left[\sum_{i,j=1}^{n_K}\bar{\varphi}(W_i;\psi_0,\hat{\eta}(I_k^c))\cdot\bar{\varphi}(W_j;\psi_0,\eta(I_k^c))\,|\,I_k^c\right]$$
$$+\frac{1}{n}\mathbb{E}\left[\sum_{i,j=1}^{n_K}\bar{\varphi}_0(W_i;\psi_0,\eta(Z_i))\cdot\bar{\varphi}_0(W_j;\psi_0,\hat{\eta}(Z_j))\right]$$
$$- 2\mathbb{E}\left[(\mathbb{G}_{n,I_k^c}\varphi(W;\psi_0,\hat{\eta}(I_k^c))\,|\,I_k^c]\cdot\mathbb{E}\left[(\mathbb{G}_{n,I_k^c}\varphi_0(W;\psi_0,\eta(Z)))\right]\right] + o_{\mathbb{P}_n}(1)$$
$$=\frac{1}{n}\sum_{i,j=1}^{n_K}o(\frac{1}{n}) + \frac{1}{n}\sum_{i,j=1}^{n_K}\mathbb{E}\left[\bar{\varphi}_0(W_i;\psi_0,\eta(Z_i))\right]\cdot\mathbb{E}\left[\bar{\varphi}_0(W_j;\psi_0,\hat{\eta}(Z_j))\right] + o_{\mathbb{P}_n}(1)$$
$$=o_{\mathbb{P}_n}(1)$$

459     The second equality is due to Assumption 5, the independence of $W_i$'s, and (A.4).

460     By Lemma B.1 of Chernozhukov *et al.* [Che+17b],

$$\mathbb{E}\left[\left[\sqrt{n_K}(\check{\psi}_k - \psi_0) - \mathbb{G}_{n,I_k^c}\varphi_0(W;\psi_0,\eta(Z))\right]^2 \mid I_k^c\right] = o_{\mathbb{P}_n}(1)$$

461     implies

$$\sqrt{n_K}(\check{\psi}_k - \psi_0) - \mathbb{G}_{n,I_k^c}\varphi_0(W;\psi_0,\eta(Z)) = o_{\mathbb{P}_n}(1)$$

462     Therefore, we have

$$\sigma^{-1}\sqrt{n_K}(\check{\psi}_k - \psi_0)_{k=1}^K = \sigma^{-1}\mathbb{G}_{n,I_k^c}\varphi_0(W;\psi_0,\eta(Z))_{k=1}^K + o_{\mathbb{P}_n}(1) \overset{d}{\to} (\mathcal{N}_k)_{k=1}^K$$

463     where $(\mathcal{N}_k)_{k=1}^K$ is a Gaussian vector with independent $\mathcal{N}(0,1)$ coordinates. Using the independence
464     of $Z_i$'s and the central limit theorem, we have

$$\sigma^{-1}\sqrt{n}(\tilde{\psi} - \psi_0)$$

$$= \sigma^{-1}\sqrt{n}\left(\frac{1}{K}\sum_{k=1}^K(\check{\psi}_k - \psi_0)\right)$$

$$= \frac{1}{K}\sigma^{-1}\sum_{k=1}^K \mathbb{G}_{n,I_k^c}\varphi_0(W;\psi_0,\eta(Z)) + o_{\mathbb{P}_n}(1)$$

$$\overset{d}{\to} \frac{1}{K}\sum_{k=1}^K \mathcal{N}_k = \mathcal{N}(0,1).$$

465     **Step 2:** This step demonstrates (A.2). Observe that for some constant $C_\varepsilon$ that depends only on $\varepsilon$ and $\mathcal{P}$,

$$||\varphi(W;\psi_0,\hat{\eta}(I_k^c)) - \varphi(W;\psi_0,\eta(Z))||_{\mathbb{P}_n,2} \le C_\varepsilon(\mathcal{I}_1 + \mathcal{I}_2 + \mathcal{I}_3),$$

466     where

$$\mathcal{I}_1 = \max_{d\in\{0,1\}}||\tilde{Q}_n(d,Z;\hat{\gamma}_n^{Q,I_k^c}) - Q(d,Z)||_{\mathbb{P}_n,2},$$

$$\mathcal{I}_2 = ||\frac{T(Y - \tilde{Q}_n(1,\lambda;\hat{\gamma}_n^{Q,I_k^c}))}{\tilde{g}_n(\cdot;\hat{\gamma}_n^{g,I_k^c})} - \frac{T(Y - Q(1,Z))}{g(\lambda)}||_{\mathbb{P}_n,2},$$

$$\mathcal{I}_3 = ||\frac{(1-T)(Y - \tilde{Q}_n(0,\lambda;\hat{\gamma}_n^{Q,I_k^c}))}{1 - \tilde{g}_n(\cdot;\hat{\gamma}_n^{g,I_k^c})} - \frac{(1-T)(Y - Q(0,Z))}{1 - g(\lambda)}||_{\mathbb{P}_n,2},$$

467     We bound $\mathcal{I}_1, \mathcal{I}_2$, and $\mathcal{I}_3$ in turn. First, $\mathbb{P}_n(\mathcal{I}_1 > \delta_{n_K}) \le \Delta_{n_K} \to 0$ by Assumption 4, and so
468     $\mathcal{I}_1 = o_{\mathbb{P}_n}(1)$. Also, on the event that

$$\mathbb{P}_n(\varepsilon \le \tilde{g}_n(Z;I_k^c) \le 1 - \varepsilon) = 1 \tag{A.5}$$

$$||\tilde{Q}_n(1,\lambda;\hat{\gamma}_n^{Q,I_k^c}) - Q(1,Z)||_{\mathbb{P}_n,2} + ||\tilde{g}_n(\cdot;\hat{\gamma}_n^{g,I_k^c}) - g(Z)||_{\mathbb{P}_n,2} \le \delta_{n_K}, \tag{A.6}$$

469     which happens with $P_{\mathbb{P}_n}$-probability at least $1 - \Delta_{n_K}$ by Assumption 4,

$$\mathcal{I}_2 \le \varepsilon^{-2}||Tg(Z)(Y - \tilde{Q}_n(1,\lambda;\hat{\gamma}_n^{Q,I_k^c})) - T\tilde{g}_n(Z;I_k^c)(Y - Q(1,Z))||_{\mathbb{P}_n,2}$$

$$\le \varepsilon^{-2}||g(Z)(Q(1,Z) + \zeta - \tilde{Q}_n(1,\lambda;\hat{\gamma}_n^{Q,I_k^c})) - \tilde{g}_n(Z;I_k^c)\zeta||_{\mathbb{P}_n,2}$$

$$\le \varepsilon^{-2}||g(Z)(\tilde{Q}_n(1,\lambda;\hat{\gamma}_n^{Q,I_k^c}) - Q(1,Z))||_{\mathbb{P}_n,2} + ||(\tilde{g}_n(Z;I_k^c) - g(Z))\zeta||_{\mathbb{P}_n,2}$$

$$\le \varepsilon^{-2}||\tilde{Q}_n(1,\lambda;\hat{\gamma}_n^{Q,I_k^c}) - Q(1,Z)||_{\mathbb{P}_n,2} + \sqrt{C}||\tilde{g}_n(Z;I_k^c) - g(Z)||_{\mathbb{P}_n,2}$$

$$\le \varepsilon^{-2}(\delta_{n_K} + \sqrt{C}\delta_{n_K}) \to 0,$$

470     where the first inequality follows from (A.5) and Assumption 4, the second from the facts that
471     $T \in \{0,1\}$ and for $T = 1, Y = Q(1,Z) + \zeta$, the third from the triangle inequality, the fourth from
472     the facts that $\mathbb{P}_n(g(Z) \le 1) = 1$ and $\mathbb{P}_n(\mathbb{E}_{\mathbb{P}_n}\left[\zeta^2 \mid Z\right] \le C) = 1$ in Assumption 3, the fifth from

473  (A.6), and the last assertion follows since $\delta_{n_K} \to 0$. Hence, $\mathcal{I}_2 = o_{\mathbb{P}_n}(1)$. In addition, the same
474  argument shows that $\mathcal{I}_3 = o_{\mathbb{P}_n}(1)$, and so (A.2) follows.

475  Step 3: This step demonstrates (A.3). Observe that since $\psi_0 = \mathbb{E}_{\mathbb{P}_n}[Q(1, Z) - Q(0, Z)]$, the left-
476  hand side of (A.3) is equal to

$$\mathcal{I}_4 = \sqrt{n} \int \frac{\tilde{g}_n(Z; I_k^c) - g(z)}{\tilde{g}_n(Z; I_k^c)} \cdot (\tilde{Q}_n(1, \lambda; \hat{\gamma}_n^{Q, I_k^c}) - Q(1, z))$$
$$+ \frac{\tilde{g}_n(Z; I_k^c) - g(z)}{1 - \tilde{g}_n(Z; I_k^c)} \cdot (Q(0, z; I_k^c) - Q(0, z)) \, \mathrm{d}\mathbb{P}_n(z).$$

477  But on the event that

$$\mathbb{P}_n(\varepsilon \leq \tilde{g}_n(Z; I_k^c) \leq 1 - \varepsilon) = 1$$

478  and

$$\max_{d \in \{0,1\}} ||\tilde{Q}_n(d, \lambda; \hat{\gamma}_n^{Q, I_k^c}) - Q(d, Z)||_{\mathbb{P}_n, 2} \cdot ||\tilde{g}_n(Z; I_k^c) - g(Z)||_{\mathbb{P}_n, 2} \leq \delta_{n_K} \cdot n_K^{-1/2},$$

479  which happens with $P_{\mathbb{P}_n}$-probability at least $1 - \Delta_{n_K}$ by Assumption 4,the Cauchy-Schwarz inequality
480  implies that

$$\mathcal{I}_4 \leq \frac{2\sqrt{n}}{\varepsilon} \max_{d \in \{0,1\}} ||\tilde{Q}_n(d, \lambda; \hat{\gamma}_n^{Q, I_k^c}) - Q(d, Z)||_{\mathbb{P}_n, 2} \cdot ||\tilde{g}_n(Z; I_k^c) - g(Z)||_{\mathbb{P}_n, 2} \leq \frac{2\delta_{n_K}}{\varepsilon} \to 0,$$

481  which gives (A.3).

482  $\qquad\qquad\qquad\qquad\qquad\qquad\qquad\qquad\qquad\qquad\qquad\qquad\qquad\qquad\qquad\qquad\qquad\qquad\qquad\qquad\qquad\qquad$ $\square$