[Reviews · NeurIPS 2019]

Reviewer 1



This work considers causal inference in the presence of unobserved confounding and studies the case where a proxy is available for the unobserved confounding in the form of a network connecting the units. This is an important problem and applying embedding to causal inference seems a natural way nowadays. The novelty lies in using a network proxy to estimate the unobserved confounder. The theoretical results build upon Chernozhukov et al and modified to allow for the use of the full data in the embedding training. The paper writing seems rush to me. There are a few issues/confusion here: 1) Page 3, Section 2. Embedding methods. Anonymous [Ano] 2) Page 3, last paragraph in Section 3, what is \hat{\eta}_n? 3) Page 3, Equation in Example 4.2, what is \sigma?

Reviewer 2



Summary: The paper introduces a new method for causal effect estimation by exploiting social network structure to capture possible confounding aspects between treatment and outcome, which can subsequently be adjusted for. It translates a social network into an embedding that can be used as a proxy variables for the actual confounders. It is based on the assumption that closely linked friends in a network are likely to be similar. Performance is evaluated on partially simulated real-world data on various data sets. The subject is interesting and difficult: many tech giants are eager to exploit and extract all kinds of new information from social networks, and this application could offer a very interesting opportunity. However, despite this intriguing backdrop, the paper itself does not manage to convince at all points. The main rationale for using the (social) network is that ‘similar people are likely to be friends’ (l..21). This makes intuitive sense, however it does not imply that people that are far apart in a social network are likely to be very dissimilar. As the resulting similarity measures from the network embeddings are subsequently used as proxy for the actual confounding variables, it would be helpful to provide a slightly more in-depth example + explanation of how effective this proxy actually captures the key confounding aspects of the unobserved variables. At the moment we are required to trust in the ‘black box’ (l.136) … which is ok, but I still would like a bit more reassurance to start from. Unfortunately the current evaluations provided in section 6 are not enough due to the lack of a reliable ground-truth. I applaud the authors for making the assumptions behind their approach very specific (as every principled causal method should do). However, this also lays bare some of the potential weak points behind it. The translation of the network into a predictive embedding that serves as a good proxy for use in a causal inference procedure is interesting and promising. However, then asking a practitioner to assess the plausibility of this predictive embedded model is not realistic in practice. For a generating model s/he will at least have some notion of possible mechanisms to give a good assessment … but for such an abstract embedded model this is almost impossible unless s/he has a huge amount of experience in judging such models, as already acknowledged by the authors (l.209). However, a more important issue is that the current assumptions are either not complete or do not seem to guarantee validity. In particular: if the observed joint density of a straightforward model X <- Z -> (T -> Y), (i.e. X good proxy of confounder Z of causal link from T to Y) follows a multivariate Gaussian distribution, then the system satisfies the assumptions, and adjusting on Z (or X) will indeed lead to a good approximation of the causal effect of T on Y. However, there ALSO exists a linear Gaussian model T -> (Z -> Y) + Z -> X, (i.e. X good proxy of partial mediator Z of causal link between T and Y) that matches the exact same distribution, and for that model/system adjusting on Z or X would clearly lead to a wrong causal prediction. Essentially you require that X and Z are not causally affected by treatment T. But this is an assumption on the underlying model: if satisfied then assumption 1 automatically follows, but not necessarily the other way around. Furthermore, you start from X as a noisy proxy of the actual confounder Z, but then as stated in l.201, Assumption 2 essentially states that X captures ‘whatever information is relevant to the prediction of the treatment and outcome’. That does not correspond to a noisy observation, but essentially still relies on obtaining full information on the effect of the confounding variable. I was hoping for a bound on the accuracy of the estimated causal effect in relation to the ‘closeness’ of the proxy variable X to the actual confounder Z. The hope expressed in l.210-213 only holds if both observed/proxy and unobserved/exogenous confounding have the same sign and the second is not stronger than the first, otherwise the unadjusted estimate may actually be closer to the ground truth. (This also holds for the experiment behind Fig.1). On a final note: I have also had the pleasure of reviewing the closely related ‘Using Text Embeddings for Causal Inference’ which conceptually seemed more interesting than this one. In conclusion: the problem is interesting and highly relevant, and the approach is promising. However, there is some concern about the actual validity in section 5, and the experiments don’t seem sufficiently rigorous to fully capture a proper evaluation of efficacy of the method. Together with the strong overlap with the other paper mentioned above I find this the weaker one, and hence recommend reject.

Reviewer 3



# Summary The authors present a method that exploits graph embedding methodology to use graph structure as a proxy in causal estimation problems. They provide sufficient conditions under which adjusting for an embedding learned without reference to a specific generative model will yield asymptotically unbiased and asymptotically normal estimates of causal effects at the parametric rate. They then propose an embedding network architecture that is designed to estimate embeddings that could plausibly satisfy these conditions. They conduct a series of experiments using semisynthetic data to demonstrate how these methods could work in practice. # Feedback This paper was a pleasure to read. The exposition is very clear, assumptions are appropriately foregrounded, and the experiments are designed to test the claims that the authors set out to make. Full disclosure: I reviewed an earlier version of this paper, and am happy to say that it is much improved. ## Elephant in the Room: Assumption 2 My main concern rests with the identifying Assumption 2. It is left unclear how one might reason about this assumption being met in practice. The authors rightly call this out as the crux of the method, and the most difficult assumption to evaluate. However, this discussion still leaves out how one might judge the “plausibility of the predictive embedding model” without falling back to judging a generative model of at least some aspect of the network. In my opinion, assumptions are only as weak as the heuristic by which they can be judged, so I think this question needs to be addressed. One suggestion: the embedding loss defines a set of sufficient statistics of the network that the embedding is designed to reconstruct. I think a condition relating lambda(Z) to these sufficient statistics (for example, a completeness condition stating that unique values of the latent variable map to unique distributions of the sufficient statistics) could be a good starting point. Completeness conditions play a major role in other work on proxy methods. This condition could probably be weakened to focus on the task of predicting treatment or outcomes, but I think there needs to be some formalization of how much confounding information is carried by the observed network. Reasoning about these conditions could also help the user design the Sample() function, which is a major degree of freedom in this approach that could also use some discussion. This being said, the exogenous confounding do a nice job of empirically probing part of this assumption to some extent. However, it doesn’t help the reader to try to reason about how confounding information might be represented in the aspects of the network that are being modeled by the embedding. ## Empirical Experiments A major strength of the paper is the empirical experiments, although, as with all experimental sections, there are more aspects to explore. I have one major concern with the statement that adjusting for the network always helps, even when confounding is not fully explained by the network. This is not generally true for confounder adjustment: it is well-known that adjusting partway for confounders that are highly predictive of treatment but not of outcome can increase bias. This phenomenon is known as bias amplification or Z-bias (see, e.g., Middleton et al https://www.cambridge.org/core/journals/political-analysis/article/bias-amplification-and-bias-unmasking/B95DDA52BE93B761C067EEE60739DDBD, Ding et al https://academic.oup.com/biomet/article/104/2/291/3737784). The simple generating process in this experiment does not admit this behavior because the propensity score enters directly into the prognostic score, so it is not possible to explain variation in the treatment without also explaining a proportional amount of variation in the outcome. I would expect to see this problem arise in an experiment where the network is highly predictive of treatment assignment and much less so of outcome, but this possibility is excluded by design in the experiment. This is not a damning concern, but I would encourage the authors to consider tempering their conclusions on this issue, and to raise the possibility in a discussion of results. Ultimately, one still needs to reason about what information is conveyed by the graph embedding. There is also some amount of control that the user has about what information the embedding will carry based on the weights they put on the two predictive portions of the network. Perhaps this could be suggested as future work. Smaller questions: Why is the parametric model given 128 blocks? I am curious about what would happen if you gave the parametric model three blocks, as I imagine at least the district latent variable being expressed as a block structure in the network. Is it possible to visualize how the user features manifest in the network? For example, does the adjacency matrix appear to have 3 communities determined by district? ## Nitpicks Miao et al does not need the marginal distribution to be identifiable. That is the primary contribution of that paper. For the sufficiency of Q(1, Z), people often cite Hansen 2008 https://academic.oup.com/biomet/article-abstract/95/2/481/230183 as well. In the network ERM objective, does the subgraph always include all of the original nodes, and are you only subsampling edges? If not, what do you do for units that are not included as vertices in the subgraph. ------------------------- # Post-Rebuttal Feedback I am still bullish on this paper because I think it presents an innovative approach to the problem and explains all of its moving parts clearly. I think somebody can read this paper and know exactly why they would dispute conclusions based on the methodology. I am a bit disappointed in the response regarding Assumption 2. I was hoping for a little bit of deeper thought on this issue (especially given that this is the second round of feedback that's focused on this question). I don't find the analogy to image models to be particularly compelling, because we can directly assess the performance of those models against ground truth, but only have our assumptions to rely on in causal inference problems. In fact, there are active debates about the reliability of decision-making that relies on the internal representations of these models. So I think the authors need to reflect a bit more and modify their advice here. I think they need to at least highlight a number of the important factors discussed in the proxies/measurement error literature that determine how well causal effects can be recovered. All of that said, I think this paper still meets the bar for publication. This would also make it easier to have discussions out in the open about the potential pitfalls of using embeddings as proxies for confounding variables, which I think would be useful for the community as a whole.

[Author Response · NeurIPS 2019]

We thank the reviewers for reading the paper and for their detailed comments. We are happy that all reviewers agree the problem is important and that the approach is natural and interesting.

**Assumption 2** R3 and R4 note that assumption 2—the embedding method extracts the salient predictive information from the network—may be difficult to assess in practice. This is true, and, as both reviewers note, we make this explicit in the paper. We say that such judgments should be based on experience with the embedding methods, and empirical performance in related situations. For instance, if a graph embedding method does well at predicting many features associated with social identity (e.g., age, music preference, political affiliation) in a number of contexts then we can expect that the embedding method will adjust for features associated with social identity. By analogy, it is commonly believed that convolutional neural nets extract semantically-meaningful representations of images, but this belief exists independent of supporting theory, generative models, or clear heuristics for reasoning about when such representations might succeed. Direct judgments about what sort of information a method might extract are often easier to make than judgments about assumptions required for (hypothetical) formal results.

That said, we agree identifying conditions that guarantee good behavior of embedding methods is an important and fascinating topic with application to the causal inference approach here, and is an good direction for future work. However, establishing such results is outside the scope of the paper. We think (and we believe the reviewers agree) that the contribution is appropriately developed for a NeurIPS paper exploring a newly-established and unconventional method.

**Partially correcting for confounding** R3 and R4 both note that partially correcting for confounding may hurt. This is indeed possible for the reasons the reviewers point out. We have added a discussion of this point and the need for the investigator to consider it. The point about variables that strongly affect treatment but not outcome is well-taken (though note that since the embedding construction extracts information relevant to both treatment and outcome, the method automatically mitigates this somewhat).

**R2** We have corrected the typos you point out (1. is anonymized ref, 2. is estimate of all nuisance params, 3. is sigmoid)

**R3** *Linear Gaussian Model* You point out that it is possible for a linear Gaussian model with the wrong (mediator) causal structure to satisfy Assumption 1. We agree; this was a minor error on our part. We have followed your suggestion and replaced assumption 1 with the (slightly stronger) assumption of the DAG such that the network and $Z$ are not affected by the treatment. Beyond the fact that it fixes the bug, we think this condition is more interpretable—thank you!

*X is not a 'noisy' observation.* We have clarified this point. This is an important way in which the network setting deviates from the iid proxy setting. In the iid setting, each $x_i$ is a noisy version of the associated $x_i$. In the network setting, we expect to get more and more information about each node as the network grows. For example, in stochastic block models, community identities can be recovered exactly under very weak conditions (Bickle and Chen 2009). Intuitively, your social identity can be fully inferred in the limit of infinite possible possible friendships. The assumption that the network asymptotically fully reveals $z$ is appropriate (or, at least, standard) in this setting.

*Text Embeddings for Causal Inference* You write that you read the text embeddings paper, and suggest merging it with the networks paper. We are gratified you enjoyed it. In our view, the papers are complementary, and should be published independently. Indeed, the first version of this work combined both ideas in a generic 'embeddings for causal inference' paper. Reviewers for ICML (including R4) encouraged us to separate the papers. That was the correct choice—each paper is much improved in clarity and development in this way. The results from each paper do not anticipate the results from the other—the fact that we can use graph embeddings to correct for unobserved confounding in networks does not make it obvious that we can use document embeddings for identification-preserving dimension reduction in text. Beyond that, the technical development is quite different: the text paper does not address non-iid data, and does not require the extension of double-ML methods to handle this. Further, the experimental setup and implementation (data, methods, and challenges) are disjoint for the two papers.

That said, if you are committed to only supporting one of the two papers, we ask that you make it the networks one. This paper handles a more unusual learning situation, and has a more substantial technical development.

(For the benefit of R4, we note that although the network paper is a small revision of the earlier version, the text paper was hugely reworked, and now focuses on dimension reduction and applications. Unfortunately, NeurIPS policy does not allow external links in reviewer response.)

**R4** Thank you for the second round of careful comments! We are gratified you agree that the paper is much improved. *Why is the parametric model given 128 blocks?* So that the dimension of the community assignment vector matches the capacity of the 128-dimensional embedding model. 128 was chosen because it is the default setting for the graph embedding method we used. We chose to match to avoid hyperparameter tuning issues, but note that (i) the network is very large, so the high capacity should not disadvantage the blockmodel, and (ii) informal testing suggested results remain poor irrespective of number of blocks.

[Meta-Review · NeurIPS 2019]

The reviewers agreed that the paper provides a novel and interesting way to approach a difficult problem: estimating causal effects in network data, under a homophily assumption. Two of the reviewers commended the clarity with which the authors approached the subject. Beyond praising the writing in general, the reviewers specifically commended the paper for being very explicit and clear about the assumptions needed for an embedding method to work in the context of causal inference in network data. One of the reviewers also noted that extending the estimation theory of double machine learning to the network, non-iid setting, is a good technical contribution. The reviewers were concerned about the following points: 1. The focus on homophily. 2. A technical point about linear-Gaussian models, which was corrected in the authors’ response. 3. The difficulty in using a black-box embedding model for adjusting. Since in causal inference one does not have a test-set, it is hard to know when is a model sufficient for obtaining the correct estimates. The proposed method relies on black-box embedding models, making it more difficult to develop confidence, or even intuition, as to how well they work. 4. Expanding on the previous point, the reviewers were concerned about Assumption 2: the assumption that the embedding model indeed captures all the confounding information. Specifically, they were concerned that this will be even harder to argue for compared with standard causal inference tasks. In standard causal inference, one hopes to adjust for all confounders directly. Knowing what are the confounders is mainly done via domain knowledge and statistical tests which might help gain intuition, though they can never be conclusive. The concern here is that using an embedding adds another layer of complexity to this process, and that there are insufficient proposals, even on the heuristic level, on how a practitioner could go about assessing the validity of this assumption and the strength of the embedding model in capturing confounding. Regarding point 1: I think focusing on homophily is 100% fine for a paper which is dealing with a new and difficult subject. Point 2 was fixed in the response. Regarding points 3 and 4: I strongly agree with Reviewer 4 here. I think this is a very difficult problem, and the solution the authors propose seems like one of the most promising directions currently available. As such, the paper is commendable for laying bare the crux of the difficulty, for which I do not see any immediate easy fix. The next step in my eyes is to have the discussion in the open community. The paper is a first step, offering both a method and a set of carefully constructed benchmarks. I expect other work to follow up on this, going deeper into what can be done to evaluate and gain confidence in embedding models as proxies to confounders.